# Monitoring of Chemical Changes in Coffee Beans during the Roasting Process Using Different Roasting Technologies with Nuclear Magnetic Resonance Spectroscopy

**Vera Gottstein [1], Katrin Krumbügel [1], Thomas Kuballa [1], Steffen Schwarz [2] , Enrico Walch [3], Pascal Walch [3] and Dirk W. Lachenmeier [1,\*]**

[1] Chemisches und Veterinäruntersuchungsamt (CVUA) Karlsruhe, Weissenburger Strasse 3, 76187 Karlsruhe, Germany; vera.gottstein@gmail.com (V.G.); katrin.krumbuegel@hotmail.de (K.K.); thomas.kuballa@cvuaka.bwl.de (T.K.)

[2] Coffee Consulate, Hans-Thoma-Strasse 20, 68163 Mannheim, Germany; schwarz@coffee-consulate.com

[3] Kammerer GmbH, An der B10, 75196 Remchingen, Germany; e.walch@tyboon.com (E.W.); p.walch@tyboon.com (P.W.)

\* Correspondence: lachenmeier@web.de; Tel.: +49-721-926-5434

**Abstract:** The roasting process is an important step in coffee production, leading to important physical and chemical changes that are responsible for the sensory quality of a coffee beverage. Besides the commonly used drum roasters, a newly developed infrared roaster can be used to roast green coffee beans. In this study, [1]H nuclear magnetic resonance (NMR) spectroscopy was used to analyze the fat and aqueous extracts of coffee beans roasted to different degrees of roasting using a professional drum roaster, a hot air fluidized bed sample roaster and an infrared roaster. Caffeine-containing and decaffeinated *Coffea arabica* coffee samples were used to monitor the roasting process in the different roasters. Compared with the drum-roasted coffee sample, the formation and degradation of NMR-detectable components in the coffee sample roasted with the infrared roaster and the hot air roaster were time-dependent. In the decaffeinated coffee sample, compounds such as kahweol, caffeoylquinic acid and trigonelline were found to occur at lower levels. The formation and degradation of the NMR-detectable compounds in the decaffeinated coffee sample also occurred with a time lag or to a lesser extent than in the caffeine-containing coffee sample.

**Keywords:** roasting process; coffee; decaffeinated coffee; NMR; drum roaster; infrared roaster; hot air roaster

## 1. Introduction

As a beverage, coffee is a popular stimulant around the world. There are specific coffee creations that have become an important part of culture over the years. Global coffee consumption has been growing steadily since 1990, and the International Coffee Organization reports that global consumption exceeded 167 million bags (1 bag = 60 kg) in 2020/2021 [1]. Of the 129 existing *Coffea* species, *C. arabica* and *C. canephora* are of the highest economic importance. Several steps are required to consume coffee as a beverage, including the harvesting of coffee cherries, the processing of green coffee, the roasting process and the brewing. Coffee roasting is a dry and fat-free heating process that has a major impact on the chemical composition of coffee. Heat can be transferred from the roaster to the coffee beans via convection (through contact with hot air), conduction (through direct contact with a hot surface) and radiation (such as in the infrared spectral range) [2–4]. The most commonly used roaster is the drum roaster, in which the coffee beans are heated by hot air in a rotating drum. Heat is also transferred by the beans' contact with the hot surface of the roaster, and thus, this method involves a mixture of convection and conduction [4]. Another roasting method is fluidized bed roasting, in which hot gas is directed at the beans at high velocity. In this way, the beans are both moved and heated by the gas phase [3,5].

The coffee-roasting process is an important step in which the coffee beans undergo various chemical and physical changes. There is the formation and degradation of various compounds that are important for the organoleptic quality characteristics of roasted coffee [6]. More than 300 compounds were identified in roasted coffee, including various volatile compounds, melanoidins, organic acids (lactic acid, acetic acid, formic acid, malic acid, citric acid, glycolic acid), bioactive compounds (caffeine, trigonelline, *N*-methylpyridine (NMP), phenolic compounds (chlorogenic acids), lipids (triacylglycerides (TAG), fatty acids), diterpenes (kahweol, cafestol) and heat-induced contaminants (5-hydroxymethylfurfural (HMF), acrylamide, furfuryl alcohol) [6–8]. In addition to organoleptic and quality properties, some of these compounds are associated with positive and negative effects on human health. Caffeine is one of the most widely consumed psychoactive substances [9]. Trigonelline and its thermal degradation product NMP are both nitrogenous compounds [9–11]. They may have some beneficial effects on cellular energy metabolism and chemopreventive and antioxidant activities [9–11]. Chlorogenic acids show anti-inflammatory and neuroprotective activities and are antioxidants [12]. Formic acid, acetic acid and lactic acid contribute to the acidic taste of coffee. Studies indicate that the diterpenes cafestol and especially kahweol may have antioxidant, anti-tumor, chemoprotective and anti-inflammatory effects [13]. In contrast, HMF and furfuryl alcohol are components of concern in roasted coffee. The heat-induced contaminant furfuryl alcohol has been classified as a possible human carcinogen by the International Agency for Research and Cancer (IARC) [14], and studies reported some evidence of carcinogenic activity of HMF in animal experiments [15]. However, the final composition of roasted coffee is influenced by the *Coffea* species, variety, geographic origin, cherry processing and roasting process [6]. It is also possible to remove caffeine via a decaffeination process using solvents, such as dichloromethane, water, ethyl acetate or carbon dioxide, prior to roasting [4].

To improve the quality of roasted coffee and minimize heat-induced contaminants, analytical methods for monitoring the roasting process are useful in addition to cupping. Several studies described analytical methods for monitoring the roasting process, including near-infrared spectroscopy, nuclear magnetic resonance (NMR) spectroscopy, color analysis, high-performance liquid chromatography with a diode array detector (HPLC-DAD) and gas chromatography–mass spectrometry (GC-MS) [3,5,7,16–24]. However, the monitoring of the roasting process is mainly done when using a drum roaster. In this study, the roasting process was monitored in a drum roaster, a sample roaster based on hot air roasting (hot air roaster) and a recently developed infrared roaster, which differ greatly in the mechanisms of heat transfer. In contrast with the drum roaster, heat transfer in the infrared roaster is in the form of radiation [25], whereas heat transfer in the sample roaster is in the form of convection. Various analytical techniques, including moisture content and NMR spectroscopy, were used to monitor the roasting process in the three roasters. Since NMR is a primary analytical technique, the simultaneous quantification of multiple components with simple sample preparation makes this method suitable for monitoring the roasting process [26,27]. The objective was to monitor the physical changes, such as moisture content, and chemical changes of NMR-detectable coffee components as a function of the roaster used and the degree of roasting. Both aqueous and fat extracts of roasted coffee samples were used to study the chemical changes in NMR-detectable coffee components.

## 2. Materials and Methods

### 2.1. Samples

*C. arabica* var. Catuai was used for the experiments. The first Catuai (C1) was from Fazendas Dutra (São João do Manhuaçu, MG, Brazil, coordinates: 20°18′49.7″ S, 42°07′33.9″ W), where the coffee cherries were processed using the pulped natural method. The second Catuai (C2) was from Finca Hamburgo (Chiapas, Mexico, coordinates: 15°10′24.0″ N, 92°19′46.1″ W), where the coffee was processed using the fully washed method. In addition, this coffee sample was decaffeinated using water as a solvent. During this process, the

coffee beans were exposed to water and steam to start the extraction process and expand the beans. The coffee beans were then rinsed with water to extract the caffeine. This solvent also extracts water-soluble compounds, along with caffeine. The next step was to remove the caffeine from the solution using activated carbon. The caffeine beans were then allowed to reabsorb the molecules lost during the extraction process (Demus Spa, Trieste, Italy). Green coffee samples were stored in a dry place at room temperature. After the roasting process, the roasted coffee beans were packaged with an aroma protection valve (Weber Packaging GmbH, Güglingen, Germany) and stored in a dry place at room temperature.

### 2.2. Chemicals and Reagents

The reagents and chemicals were analytical or HPLC grade. Deuterium oxide (99.9 atom% D) was obtained from Deutero (Kastellaun, Germany). Deuterated chloroform-$d_1$ ($\geq$99.8 atom% D), HMF ($\geq$97%), 5-caffeoylquinic acid (5CQA) ($\geq$97%), sodium acetate ($\geq$98.5%), tetramethylsilane (TMS) and trigonelline hydrochloride ($\geq$97.5%) were purchased from Carl Roth (Karlsruhe, Germany). Citric acid monohydrate ($\geq$99.5%), orthophosphoric acid (85%) and sodium dihydrogen phosphate monohydrate ($\geq$99%) were purchased from Merck (Darmstadt, Germany). Sodium formate (99%), sodium lactate (98%), NMP iodide ($\geq$97%), 1,2,4,5-tetrachloro-3-nitrobenzene (certified reference material), ethylbenzene (99.8%, anhydrous) and 3-(trimethylsilyl)-propionic acid-$d_4$ sodium salt (TSP) (98 atom% D) were purchased from Sigma-Aldrich (Steinheim, Germany).

The buffer was prepared by dissolving 138 g sodium dihydrogen phosphate monohydrate in 1000 mL $H_2O$ and adjusting the pH to 6.0 with orthophosphoric acid.

The reference solution for the calculation of components in fat extracts consisted of the standards tetrachloronitrobenzene (5.01 g/L) and ethylbenzene (4.57 g/L) in $CDCl_3$ containing 0.1% TMS (*v/v*).

The reference solution for the calculation of the components in the aqueous extracts consisted of the standards citric acid (3.30 g/L) and lactic acid (0.720 g/L) in $H_2O/D_2O$ (9:1, *v/v*) containing TSP (0.9 g/L).

### 2.3. Roasting Process

In order to study the changes in NMR-detectable components as a function of the roasting time, 400 g of coffee (30 g of coffee for the hot air sample roaster) was roasted for 14 min, with samples taken after every minute, resulting in 14 samples with increasing roasting degree. This roasting experiment was performed for both the caffeine-containing and decaffeinated Catuai coffee samples in a drum roaster (Solar Shop Roaster, Coffee-tech Engineering, Moshav Mazliach, Israel), an infrared roaster (Tyboon 3000, Kammerer GmbH, Remchingen, Germany) and a hot air roaster (Ikawa pro sample roaster, V2-Pro, IKAWA Ltd., London, UK). The roasting profile used for this experiment is shown in Figure 1. The roasting profile was the same for all three roasters, making the results comparable.

### 2.4. Methods

#### 2.4.1. Moisture Content

The coffee samples were ground to a particle size of 0.3 mm using a coffee mill (Mahlkönig EK43, Hemro International AG, Zurich, Switzerland). The moisture content was determined automatically by analyzing 5.0 g of the ground coffee sample with a moisture analyzer (HC103, Mettler Toledo GmbH, Giessen, Germany).

#### 2.4.2. Sample Preparation for NMR Spectroscopy

Sample preparation for the analysis of coffee fat extracts using $^1$H-NMR spectroscopy was performed according to Okaru et al. [28]. Briefly, 200 mg of ground coffee samples were shaken (Combination shaker KL2, Edmund Bühler GmbH, Bodelshausen, Germany) with 1.5 mL $CDCl_3$ containing 0.1% TMS (*v/v*) for 20 min at room temperature. The extracts were then membrane filtered (Chromafil Xtra PET, 0.25 mm, 0.45 μm, Machery-Nagel GmbH & Co. KG, Düren, Germany) and 600 μL of the filtrates were filled into NMR tubes

(DeuteroQuant, O.D. 4.966 ± 0.004 mm; I.D. 4.166 ± 0.004 mm, length 17.78 cm, Deutero, Kastellaun, Germany).

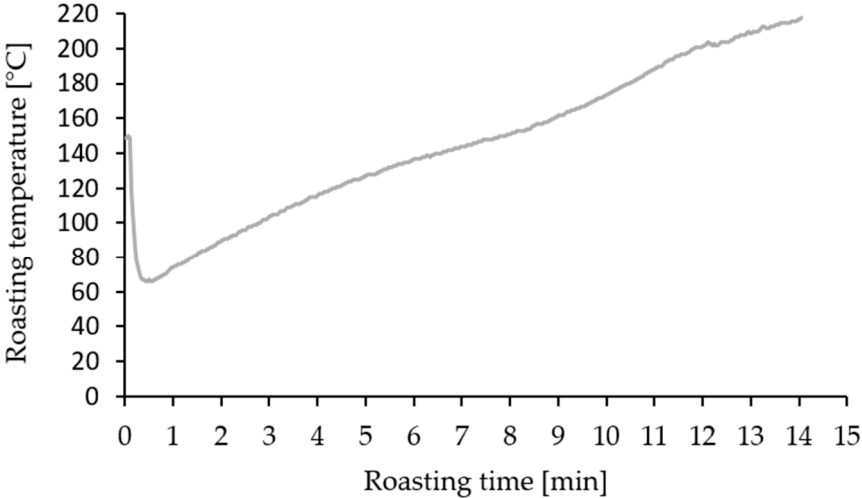

**Figure 1.** Time–temperature profile used for roasting the caffeine-containing coffee sample in the drum roaster. The same profile was used for roasting the caffeine-containing and decaffeinated coffee samples in the drum roaster, infrared roaster and hot air roaster.

For the analysis of the aqueous extracts, 600 mg of the ground coffee samples were mixed with 4 mL $H_2O$ and shaken for 20 min at room temperature (MultiReax, Heidolph, Schwabach, Germany). The extracts were filtered through a membrane filter (Chromafil GF-PET, 0.25 mm, 0.25 µm, MACHERY-NAGEL GmbH & Co., KG). An aliquot of 600 µL was mixed with a 100 µL buffer and 70 µL $D_2O$ containing TSP (10 g/L). For the [1]H-NMR measurement, 600 µL of this solution was poured into NMR tubes (DeuteroQuant, O.D. 4.966 ± 0.004 mm; I.D. 4.166 ± 0.004 mm, length 17.78 cm, Deutero).

2.4.3. NMR Spectroscopy

[1]H-NMR spectra were recorded using a Bruker 400 MHz Avance III UltraShield spectrometer with a 5 mm PASEI 1H/D-13C Z-GRD probe and a SampleXpress H15040-01 autosampler (Bruker BioSpin GmbH, Rheinstetten, Germany). All samples were temperature equilibrated for 5 min. Spectra were recorded at 300.0 K using a baseopt mode and the pulse width was automatically estimated for each sample. Spectra were processed using an exponential window function with a line-broadening factor of 0.3, followed by Fourier transformation and automatic phase and baseline correction (Topspin version 3.2, Bruker Biospin). The spectra were aligned with the TMS or TSP signals at $\delta_H = 0.00$ ppm. To ensure the quality of the spectra, the full width at half maximum values of the TMS and TSP signals were determined. A limit of 1.2 Hz was set; if this was exceeded, the measurement or sample preparation was repeated.

The [1]H-NMR measurement of coffee fat extracts was performed as described by Okaru et al. using a standard Bruker pulse program (zg30) [28]. The acquisition parameters were as follows: 128 k time domain data points, 64 scans, 2 dummy scans, 20.5503 ppm spectral width, 7.97 s acquisition time, 30 s relaxation delay (D1), 45.2 receiver gain and 300.0 K temperature. The size of the real spectrum was set to 262,144. All spectra of the coffee fat extracts were recorded with the same parameters and under the same conditions.

[1]H-NMR spectra of aqueous extracts were acquired using an optimized 1D noesygppr1d pulse sequence with water suppression and a D7 delay. The NMR spectra were acquired with 64 k time domain data points, 32 scans, 4 dummy scans, a spectral width of 20.5503 ppm, an acquisition time of 3.98 s and a receiver gain of 16. The relaxation delays D1 and D7 were 4 s and 5 s, respectively. For processing, the size of the real spectrum

was set to 131,072. All spectra of the aqueous coffee extracts were recorded with the same parameters and under the same conditions.

### 2.4.4. NMR Quantification

The concentrations of NMR-detectable coffee compounds were determined according to the PULCON principle as described by Monakhova et al. [29,30]. First, a so-called *ERETIC* factor was determined, which correlates the intensities of the signals in two separately measured solutions (reference solution containing the external standard and sample solution). The *ERETIC* factor was calculated using the following equation:

$$ERETIC = \frac{I_{Ref} \times SW_{Ref} \times MW_{Ref}}{SI_{Ref} \times c_{Ref} \times N_{Ref} \times DF} \tag{1}$$

$I_{Ref}$ is the absolute integral (*Ref*—reference substance), $SW_{Ref}$ is the spectral width, $MW_{Ref}$ is the molecular weight of the standard substance, $c_{Ref}$ is the concentration of the standard substance, $N_{Ref}$ is the number of protons producing the selected signal and *DF* is the dilution factor (to calculate the fat-soluble components: 1; to calculate the water-soluble components: 0.78). $SI_{Ref}$ is the size of the real spectrum, which is the number of data points after the Fourier transformation (256 k). The average *ERETIC* factor was used to quantify the analyte concentration using the following equation:

$$c_X = \frac{I_X \times SW_X \times MW_X \times P1_X \times V}{SI_X \times ERETIC \times N_X \times P1_{Ref} \times DF \times GW} \tag{2}$$

$I_x$ is the absolute integral (*x*—analyte), $SW_x$ is the spectral width (20.5617 ppm), $MW_X$ is the molecular weight, $SI_X$ is the size of the real spectrum (256 k), *ERETIC* is the average *ERETIC* factor of the respective standard substance (see Equation (1)), $N_X$ is the number of protons generating the selected signal and *DF* is the dilution factor (to calculate the fat-soluble components: 1; to calculate the water-soluble components: 0.78). *V* and *GW* are the volume of the extraction solution (0.004 L) and the weighed portion of the coffee sample (0.6 g), respectively. $P1_X$ and $P1_{Ref}$ are the 90° pulse widths for the reference solution and the sample, respectively.

The concentration calculation was performed automatically using MATLAB 2019b software (The Math Works, Natick, MA, USA). The routine included importing the $^1$H-NMR spectra, extracting the data points, a baseline correction, integration, quantitation according to the PULCON principle and reporting the results as Excel files. For overlapping signals of lactic acid, acetic acid, caffeine and CQA, a line-shape-fitting algorithm as described by Soininen et al. and Teipel et al. [31,32] was used for the integration.

## 3. Results and Discussion

### 3.1. Moisture Content

Table 1 shows the moisture content of grained coffee samples roasted with the drum roaster, infrared roaster and hot air roaster at different roasting times. In the green coffee beans, the moisture contents of the caffeine-containing coffee sample and the decaffeinated coffee sample were 9.8% and 9.4%, respectively. In the caffeine-containing coffee sample, a greater decrease in moisture content was observed for all three roasters until a roasting time of 10 min. The decrease in moisture content as a function of roasting time was very similar for the drum roaster and the hot air roaster in the caffeine-containing coffee sample. However, the moisture content of the caffeine-containing coffee sample roasted in the hot air roaster was slightly higher than that of the sample roasted in the drum roaster. The decrease in moisture content of the caffeine-containing coffee sample roasted in the infrared roaster was not as great as that of the samples roasted in the other two roasters. The moisture loss in the decaffeinated coffee sample was not as steep, lasting until 13 min of roasting. The progression of moisture loss in the decaffeinated coffee sample was similar for the drum roaster, infrared roaster and hot air coaster. After 14 min of roasting, the

moisture contents of the caffeine-containing and decaffeinated coffee samples roasted in the three roasters were similar, with the caffeine-containing coffee sample roasted in the drum roaster having the lowest moisture content (1.9%) and the decaffeinated coffee sample roasted in the hot air roaster having the highest moisture content (3.7%).

**Table 1.** Evolution of moisture content during the roasting process of the caffeine-containing coffee samples and decaffeinated coffee samples in the drum roaster (DR), infrared roaster (IR) and hot air roaster (HR).

| Roasting Time [min] | Caffeine-Containing Coffee Sample | | | Decaffeinated Coffee Sample | | |
| --- | --- | --- | --- | --- | --- | --- |
| | DR [%] | IR [%] | HR [%] | DR [%] | IR [%] | HR [%] |
| 0 | 9.8 | 9.8 | 9.8 | 9.4 | 9.4 | 9.4 |
| 1 | 9.4 | 8.9 | 9.5 | 8.6 | 9.2 | 9.2 |
| 2 | 8.9 | 8.4 | 9.1 | 8.5 | 8.6 | 8.9 |
| 3 | 8.3 | 8.1 | 8.6 | 8.1 | 8.0 | 8.3 |
| 4 | 7.6 | 7.6 | 7.6 | 7.5 | 7.2 | 8.1 |
| 5 | 7.2 | 6.7 | 6.7 | 6.9 | 6.9 | 7.5 |
| 6 | 5.9 | 6.6 | 6.7 | 6.5 | 6.4 | 7.6 |
| 7 | 5.3 | 5.6 | 5.4 | 6.2 | 5.8 | 6.8 |
| 8 | 4.2 | 4.8 | 4.6 | 6.0 | 5.7 | 6.6 |
| 9 | 3.4 | 4.3 | 3.9 | 5.5 | 5.1 | 6.1 |
| 10 | 2.4 | 3.6 | 2.7 | 5.3 | 4.9 | 5.5 |
| 11 | 2.6 | 4.0 | 3.2 | 4.5 | 4.3 | 4.7 |
| 12 | 2.8 | 2.6 | 3.2 | 4.4 | 4.1 | 4.5 |
| 13 | 2.3 | 3.0 | 2.8 | 3.2 | 3.0 | 3.3 |
| 14 | 1.9 | 3.0 | 2.8 | 2.7 | 3.2 | 3.7 |

*3.2. NMR Analysis of Coffee Fat Extracts*

Figure 2 shows the $^1$H-NMR spectrum of the fat extract of the caffeine-containing coffee sample (roasted for 14 min in a drum roaster). In the $^1$H-NMR spectrum, the doublet at $\delta_H$ = 5.89 ppm and the multiplet at $\delta_H$ = 7.40 ppm were used to quantify kahweol and furfuryl alcohol. The quantitation was performed automatically using the PULCON principle and the parameters listed in Table S1.

In the coffee fat extracts, the amounts of kahweol and furfuryl alcohol were determined and plotted as a function of roasting time (Figure 3). The quantification of these two compounds was performed according to Okaru et al. [28]. The concentrations of kahweol (Figure 3A) in the caffeine-containing and decaffeinated coffee samples as a function of roasting time were comparable between the drum and infrared roasters, whereas the coffee samples roasted in the hot air roaster showed lower amounts of this compound. The curves for the caffeine-containing and decaffeinated samples showed a slight increase in kahweol concentration. Since the content is related to the sample weight and not to the dry mass, this trend could be explained by the loss of water during the roasting process. Kahweol is known to be sensitive to heat, and during the coffee roasting process, this compound is degraded to form dehydrokahweol [33,34]. This degradation was not observed in this study because the signal used to quantify kahweol is also produced by dehydrokahweol. However, the amount of kahweol in the decaffeinated coffee sample was considerably lower than the amount of kahweol in the caffeine-containing sample. This observation may have been due to the decaffeination process. Water was used as the extraction solvent for caffeine removal. Water as a solvent also dissolves other components from the coffee bean that may not have been fully reabsorbed by the coffee beans after the extraction process.

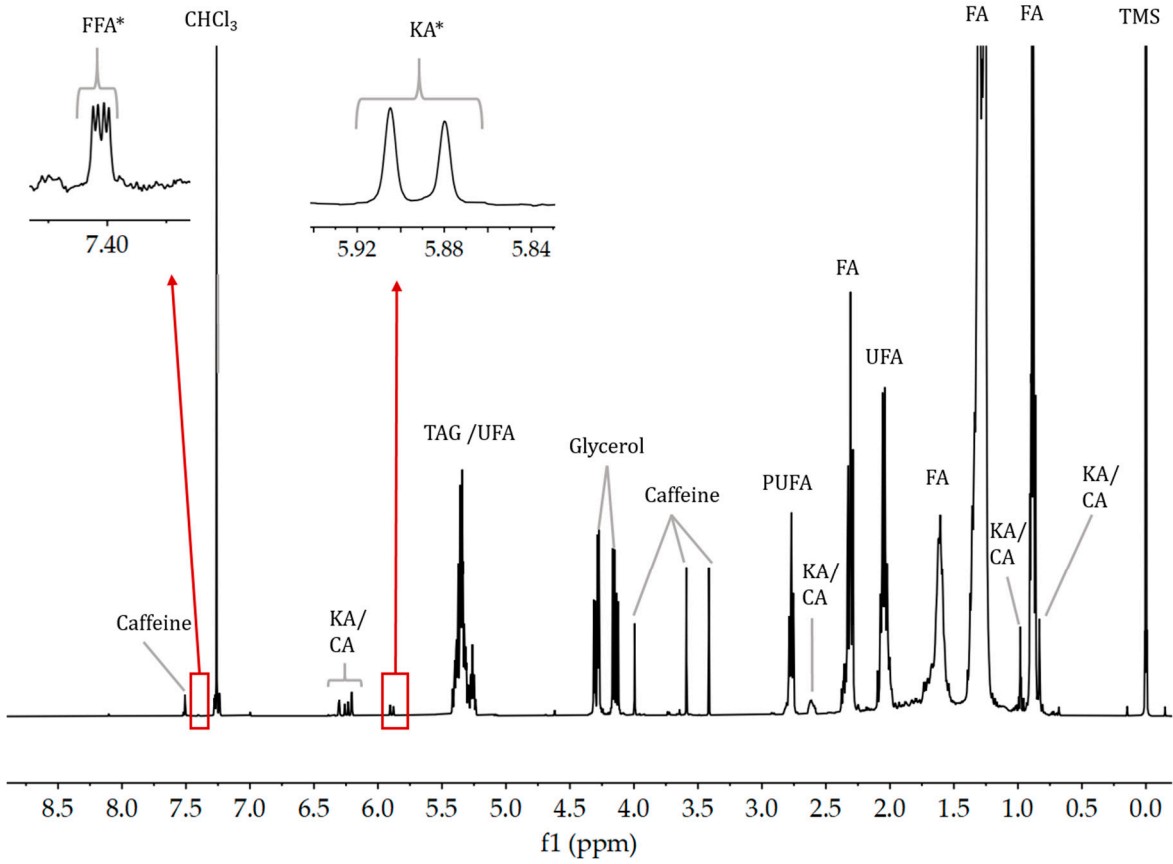

**Figure 2.** $^1$H-NMR spectrum of the fat extract of the caffeine-containing coffee sample roasted for 14 min in a drum roaster, recorded in CDCl$_3$ using a 400 MHz NMR spectrometer. Chemical shifts were calibrated using the TMS signal at $\delta_H = 0.00$ ppm. (*) KA and FFA signals used for quantification are magnified. TMS, tetramethylsilane; KA, kahweol; CA, cafestol; FA, fatty acids; UFA, unsaturated fatty acids; PUFA, polyunsaturated fatty acids; TAG, triacylglycerides; FFA, furfuryl alcohol.

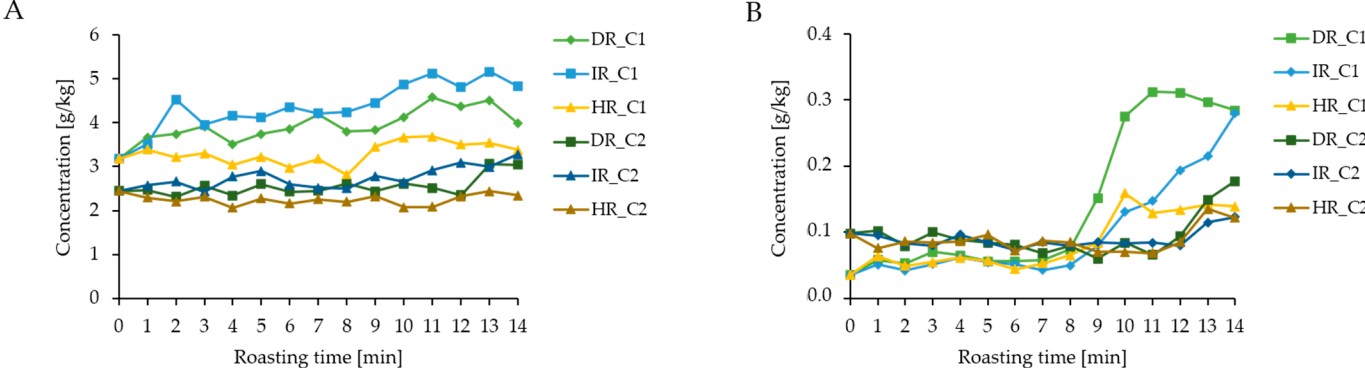

**Figure 3.** Evolution of (**A**) kahweol and (**B**) furfuryl alcohol content during the roasting process of the caffeine-containing coffee samples (C1) and decaffeinated coffee samples (C2) in the drum roaster (DR), infrared roaster (IR) and hot air roaster (HR).

For the furfuryl alcohol content (Figure 3B), the curve of the infrared roaster showed a small time lag. In particular, for the caffeine-containing coffee sample, an increase in the furfuryl alcohol content was observed for both roasters from a roasting time of 8 min onward. Compared with the infrared roaster, the concentration of furfuryl alcohol in the caffeine-containing coffee sample increased more steeply and decreased again after 10 min. Albouchi and Murkovic investigated the formation of furfuryl alcohol in two *C. arabica*

and one *C. canephora* samples as a function of time and temperature during the roasting process [35]. They also described a furfuryl alcohol concentration that reached a maximum and decreased with further roasting and explained this in terms of the polymerization of this substance to form dimers and higher oligomers [35]. In contrast, Lachenmeier et al. found the highest levels of furfuryl alcohol in the strongest roasted coffee samples [7]. In the decaffeinated coffee sample, a steady increase in furfuryl alcohol concentration was observed in the infrared roaster starting at 8 min of roasting time. In the decaffeinated coffee sample, an increase in furfuryl alcohol concentration was also observed in the drum and infrared roasters starting at 11 min of roasting time. The results indicate a lower increase in furfuryl alcohol during the roasting process in the decaffeinated coffee sample than in the caffeine-containing coffee sample. Since furfuryl alcohol is formed during the coffee-roasting process [35], the lower amount of this substance in the decaffeinated coffee indicates that the precursors were present in lower amounts in the decaffeinated coffee sample. These differences could have been due to the different geographical origins. However, some precursors may have been partially removed during extraction in the decaffeination process.

### 3.3. NMR Analysis of Aqueous Coffee Extracts

A $^1$H-NMR spectrum of the aqueous extract of the caffeine-containing coffee sample (roasted for 14 min in a drum roaster) is shown in Figure 4. From the $^1$H-NMR spectra of the aqueous coffee extracts, signals of lactic acid (doublet at $\delta_H$ = 1.33 ppm), acetic acid (singlet at $\delta_H$ = 1.94 ppm), formic acid (singlet at $\delta_H$ = 8.45 ppm), NMP (triplet at $\delta_H$ = 8.54 ppm), trigonelline (singlet at $\delta_H$ = 9.12 ppm) and HMF (singlet at $\delta_H$ = 9.45 ppm) were integrated for the quantification of these compounds. Since 5-CQA is the most abundant chlorogenic acid in coffee [36], only this compound was investigated in this work. The multiplet at $\delta_H$ = 5.30 ppm was used for the quantification of 5-CQA. The parameters for the quantification of these compounds using $^1$H-NMR spectroscopy and the PULCON principle are listed in Table S2.

The amounts of lactic acid, acetic acid, 5-CQA, formic acid, NMP, trigonelline and HMF in the caffeine-containing and decaffeinated coffee samples roasted in a drum roaster, infrared roaster and hot air roaster were calculated and plotted as a function of roasting time. The changes in the organic acids formic acid and lactic acid as a function of the roaster used and the roasting time are shown in Figure 5. From roasting at more than 6 min, formic acid formation occurred in the caffeine-containing coffee samples when using all three roasters (Figure 5A). The increase in formic acid concentration was most pronounced in the caffeine-containing coffee sample roasted in the drum roaster, and a decrease in formic acid concentration occurred here from roasting > 10 min. The formic acid curve of the caffeine-containing coffee sample roasted in the hot air roaster was similar to that of the drum roaster. However, the decrease from roasting > 10 min was not as pronounced. The increase and decrease in formic acid concentration in the caffeine-containing coffee sample roasted with the infrared roaster occurred with a slight time lag compared with the drum and hot air roasters. Wei et al. roasted *C. arabica* coffee with a sample coffee roaster at 220 °C for different roasting times and analyzed the changes using $^1$H- and $^{13}$C-NMR [37]. In their experiments, an initial increase in formic acid was observed, which changed to a decrease after 5.5 min of roasting [37]. Ginz et al. also reported an initial increase followed by a decrease in formic acid [38]. They attributed the formation to the degradation of carbohydrates in the Maillard reaction and the decrease in the volatility of this acid [38]. Formic acid formation in the decaffeinated coffee sample started at 9 min of roasting time. The curve of the formic acid concentration was very similar for the decaffeinated coffee sample roasted in the drum roaster, infrared roaster and hot air roaster. There was also a decrease in formic acid from roasting at more than 13 min, which was less pronounced in the hot air roaster compared with the drum and infrared roasters. The different formic acid curves, which were observed for the caffeine-containing and decaffeinated coffee samples, could have been due to a different heat transfer, among other reasons, such

as geographical origin. During the decaffeination process, the coffee beans underwent physical changes, which were expressed by a solidified structure of the bean tissue. This could have negatively affected the heat transfer of the roaster and can explain the time lag in formic acid formation.

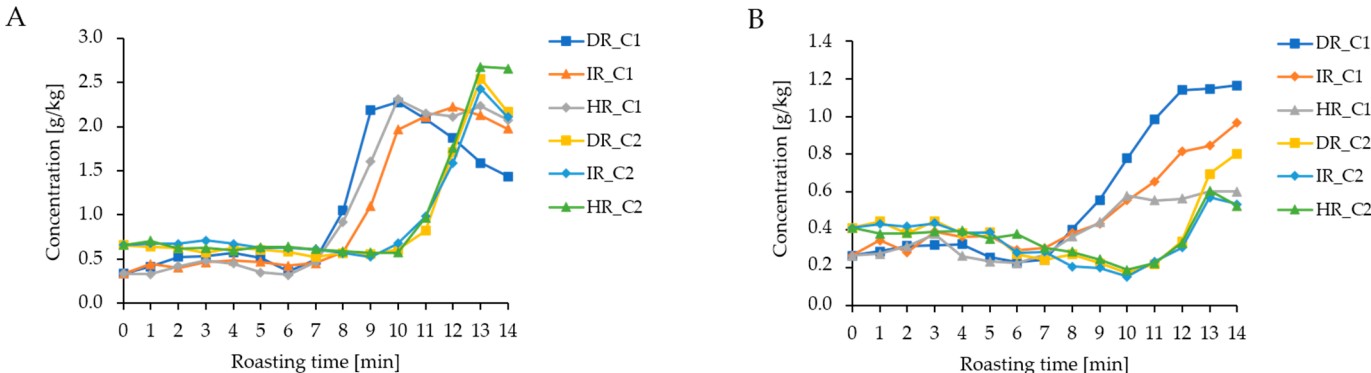

**Figure 4.** $^1$H-NMR spectrum of the aqueous extract of the caffeine-containing coffee sample roasted for 14 min in a drum roaster and recorded in $H_2O/D_2O$ (9:1, *v/v*) using a 400 MHz NMR spectrometer. Chemical shifts were calibrated using the TSP signal at $\delta_H = 0.00$ ppm. (*) Signals used for quantification. Signals of lactic acid, 5-CQA, NMR and HMF are magnified. TSP, trimethylsilylpropionic acid; NMP, N-methylpyridine; 5-CQA, 5-caffeoylquinic acid; HMF, 5-hydroxymethylfurfural.

**Figure 5.** Evolution of (**A**) formic acid and (**B**) lactic acid contents during the roasting process of the caffeine-containing coffee samples (C1) and decaffeinated coffee samples (C2) in the drum roaster (DR), infrared roaster (IR) and hot air roaster (HR).

Like the formic acid concentration curve of the decaffeinated coffee sample, the lactic acid concentration curve (Figure 5B) of this coffee sample as a function of roasting time was very similar for all three roasters. Initially, there was a slight decrease in lactic acid, followed by an increase from 10 min of roasting time onward. While the amount of lactic acid in the decaffeinated coffee sample roasted in the drum roaster continued to increase, it decreased from 13 min of roasting time onward in the sample roasted in the infrared and hot air roaster. For the caffeine-containing coffee sample, lactic acid formation started earlier at 6 min of roasting time. In the caffeine-containing coffee sample roasted in the drum roaster, an increase in lactic acid concentration was observed until 12 min of roasting time and then remained constant. In the caffeine-containing coffee sample roasted in the hot air roaster, there was only a slight increase in lactic acid formation and the concentration remained constant after 10 min of roasting. However, the concentration of lactic acid in the caffeine-containing coffee sample roasted in the hot air roaster was half that of the sample roasted in the drum roaster after 14 min. As indicated by the previously described curves, the curve of the lactic acid concentration in the infrared-roasted caffeine-containing coffee sample also showed a time lag compared with the curve of the drum-roasted sample. Wei et al. reported a steady increase in lactic acid in their roasting experiments starting from 2 min of roasting time [37]. However, they performed the roasting experiments until 9 min of roasting time. Since the lactic acid concentration in the caffeine-containing coffee sample roasted in the drum roaster in this study did not show any change from 12 min of roasting onward, the same may have occurred in the investigations of Wei et al. [37].

Plotting the concentration of acetic acid as a function of roasting time resulted in a curve similar to that of formic acid. In the roasted caffeine-containing coffee sample, an increase of this compound was observed after 7 min (Figure S1A). In the sample roasted in the drum roaster, a slight decrease was observed after 10 min of roasting. The curve of acetic acid concentration in the caffeine-containing coffee sample roasted in the infrared and hot air roaster showed a slight time lag compared with the sample roasted in the drum roaster. Wei et al. reported a similar curve for the integral of the acetic acid signal as a function of the roasting time [37]. In the decaffeinated coffee sample, an increase in acetic acid concentration occurred after 9 min of roasting, and no decrease was observed here. The curves for acetic acid formation were similar for the decaffeinated coffee samples roasted in the drum roaster, infrared roaster and hot air roaster. After 14 min of roasting, the highest concentration of acetic acid was observed in the decaffeinated coffee sample roasted with the drum roaster.

Observation of the concentration of 5-CQA as a function of roasting time revealed an increase in 5-CQA in the caffeine-containing coffee sample roasted in the drum roaster until 6 min of roasting time (Figure S1B). This was followed by a decrease until 10 min of roasting time. In the last minutes of roasting, the concentration increased again. As for acetic acid, a slight time lag of the curve was observed in the caffeine-containing coffee sample roasted in the infrared and hot air roasters. The curve of 5-CQA concentration versus roasting time does not correspond to the observations described in other studies [37]. However, due to different roasting profiles and the use of different roasters, the results obtained by Wei et al. [37] are not directly comparable with the results of this study. The initial increase in 5-CQA concentration observed in this study could have been due to isomerization. The isomerization of 5-CQA to 3- and 4-CQA prior to the formation of chlorogenic acid lactones is well known [18]. Dawidowicz et al. described additional isomerization reactions upon the heating of 5-CQA, forming *cis*-5-CQA, hydroxylated 5-CQA derivatives and dicaffeoylquinic acids (diCQA) [39]. The degradation of 5-CQA can be explained by the cleavage of the bond between quinic acid and hydroxycinnamic acids, incorporation into melanoidins, and degradation to volatile and non-volatile compounds [12,18,39]. Due to the described processes leading to the degradation of 5-CQA, a renewed increase in this compound, as observed in this study, is physicochemically unlikely. However, a change in signal shape and position of the multiplet used to quantify 5-CQA ($\delta_H = 5.30$ ppm) occurred after 10 min of roasting. These changes suggest that by integrating this signal from 10 min

of roasting time, compounds other than 5-CQA can be quantified. To verify this suggestion, further studies should be performed to identify this signal as a function of roasting time. The amount of 5-CQA in the green beans of the decaffeinated coffee sample was four times lower than that of the caffeine-containing coffee sample (Figure S1B). Additionally, it should be noted that the caffeine-containing and decaffeinated coffee samples were sourced from different geographic regions, which makes direct comparison challenging. The varying amounts of 5-CQA between the two coffees may also be attributed to their distinct origins. Furthermore, the decaffeination process may have resulted in a significant reduction in chlorogenic acid due to the extraction solvent used. The concentration of 5-CQA in the decaffeinated coffee sample showed a continuous decrease with increasing roasting time. This phenomenon was observed for the decaffeinated coffee samples roasted in the drum roaster, infrared roaster and hot air roaster. Farah et al. also reported a significantly lower amount of 5-CQA in the green beans of water-decaffeinated coffee samples and a continuous decrease with increasing roasting time [40].

In addition to changes in the concentration of 5-CQA with roasting time, shifts in the signals of the aromatic protons of chlorogenic acids with roasting time occurred in the $^1$H-NMR spectra of the aqueous extract (Figure S2). Signal shifts were also observed for caffeine signals as a function of the roasting time (Figure S2). Here, signal shifts toward the low field occurred with longer roasting times. In addition, the signals generated by the protons of caffeine became less intense and broader as the roasting time increased (Figure S2). This was most evident in the caffeine-containing coffee sample roasted in the drum roaster (Figure S2A). Changes in the signal position and shape of chlorogenic acids and caffeine can be attributed to the self-association of caffeine and formation of a 1:1 hydrophobically bound $\pi$-molecular complex between these two compounds [41].

Figure 6 shows the changes in trigonelline and NMP as a function of the roasting time and roaster used in the caffeine-containing and decaffeinated coffee samples. In the caffeine-containing coffee sample, a degradation of trigonelline occurred from a roasting time of 7 min onward (Figure 6A). The decrease in trigonelline was most dominant in the caffeine-containing coffee sample roasted in the drum roaster. Again, the curve of trigonelline concentration in the caffeine-containing coffee sample roasted in the infrared roaster showed a time lag compared with the curve of the sample roasted in the drum roaster. Only a slight degradation of trigonelline was observed in the caffeine-containing coffee sample roasted in the hot air roaster. Since trigonelline is degraded during the roasting process to nicotinic acid and NMP [6], it is not surprising that the concentration of NMP in the caffeine-containing coffee sample increased with the roasting time. With the most dominant degradation of trigonelline occurring in the drum-roasted caffeine-containing coffee sample, the largest amount of NMP was also formed in this sample (Figure 6B). The curve of NMP formation in the caffeine-containing coffee sample showed a time lag compared with the drum-roasted coffee sample, and the hot-air-roasted caffeine-containing coffee sample showed the lowest NMP formation. In the decaffeinated coffee sample, a highly reduced concentration of trigonelline was already found in the green coffee beans. No significant reduction in trigonelline was observed in the decaffeinated coffee sample roasted in the drum and hot air roasters. A degradation of trigonelline occurred in the drum-roasted decaffeinated coffee sample after 13 min of roasting. Due to the limited degradation of trigonelline, there was also only a slight increase in NMP concentration in the decaffeinated coffee sample after 10 min of roasting.

There was no detectable concentration of HMF in either the caffeine-containing or decaffeinated green coffee beans (Figure 7). In the drum-roasted caffeine-containing coffee sample, an increase in HMF concentration occurred from 6 min of roasting onward, reaching a maximum at 9 min of roasting. This was followed by a decrease to a non-detectable concentration at 13 min of roasting. In the infrared-roasted caffeine-containing coffee sample, the initial increase in HMF concentration started at 7 min of roasting and reached the maximum concentration at 10 min of roasting. After reaching the maximum concentration, there was also a decrease. In contrast with the drum-roasted caffeine-containing coffee

sample, the concentration of HMF was 0.1 g/kg after 14 min of roasting. The increase in HMF concentration in the hot-air-roasted caffeine-containing coffee sample was similar to that of the drum-roasted coffee sample. In contrast with the drum-roasted sample, the highest HMF concentration in the hot-air-roasted caffeine-containing coffee sample was found at 11 min of roasting time. After this point of roasting, only a slight decrease in HMF concentration was observed. In the decaffeinated coffee sample, an increase in HMF concentration occurred after 8 min of roasting. The highest concentration of HMF appeared after 13 min of roasting in the drum roaster and infrared roaster. The subsequent decrease in HMF concentration was greater for the drum-roasted decaffeinated coffee sample than for the infrared-roasted sample. Wei et al. and Murkovic and Bornik also observed an initial increase with a subsequent decrease in HMF concentration [37,42]. A steep increase in HMF concentration from 8 to 13 min of roasting was also observed for the decaffeinated coffee sample roasted in the hot air roaster. In contrast, the HMF concentration in the hot-air-roasted coffee sample continued to increase with roasting time.

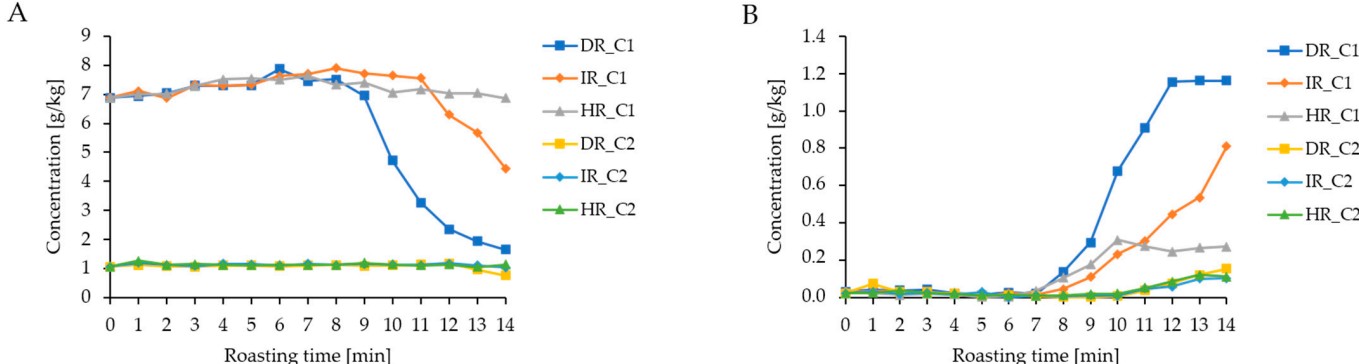

**Figure 6.** Evolution of (**A**) trigonelline and (**B**) N-methylpyridine content during the roasting process of the caffeine-containing coffee samples (C1) and decaffeinated coffee samples (C2) in the drum roaster (DR), infrared roaster (IR) and hot air roaster (HR).

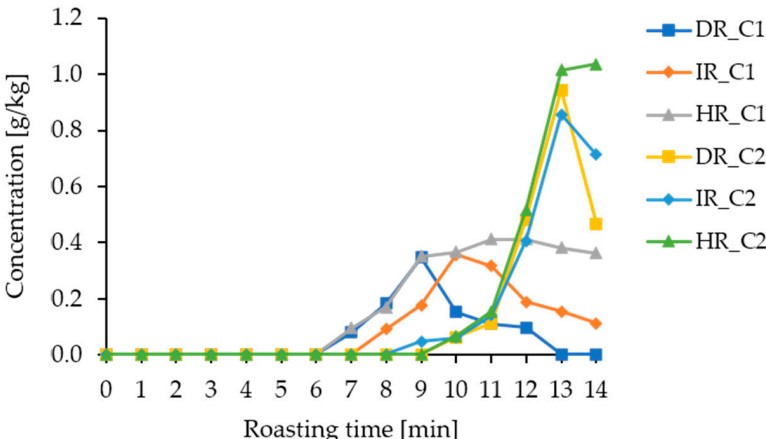

**Figure 7.** Evolution of 5-hydroxymethylfurfural content during roasting of caffeine-containing coffee samples (C1) and decaffeinated coffee samples (C2) in drum roaster (DR), infrared roaster (IR) and hot air roaster (HR).

## 4. Conclusions

In the present study, the coffee-roasting process was monitored in three different roasters with different types of heat transfer: drum roaster, infrared roaster and hot air roaster. In addition, caffeine-containing and decaffeinated coffee samples were used to monitor the roasting process. The moisture content and NMR-detectable components in the fat and aqueous extracts were analyzed to monitor the roasting process and identify

differences between the roasters and the two types of coffee samples. The changes in moisture content as a function of roasting time were similar for the coffee samples roasted in the drum roaster, infrared roaster and hot air roaster. In the caffeine-containing coffee sample, the moisture content decreased more steeply until 10 min of roasting time. In the decaffeinated coffee sample, the curve of moisture content versus roasting time decreased less steeply until 12 min. Thus, the moisture content of the caffeine-containing and decaffeinated coffee samples was similar after 14 min of roasting. The analysis of changes in the NMR-detectable compounds confirmed previous work that reported the monitoring of the roasting process of a caffeine-containing coffee sample using a sample roaster [37]. For formic acid and acetic acid, initial increases in the concentrations were observed as a function of the roasting time, followed by decreases. In contrast, the concentration of lactic acid increased continuously and the concentration of 5-CQA decreased with increasing roasting time. During roasting, trigonelline was degraded to form NMP. Thus, the concentration of trigonelline decreased while the concentration of NMP increased. The concentrations of the heat-induced contaminants furfuryl alcohol and HMF were below the limits of detection at the beginning of the roasting process. The concentrations initially increased and then decreased with further roasting.

The differences in the use of the drum roaster, infrared roaster and hot air roaster were most evident for the caffeine-containing coffee sample. There was a slight time lag in the formation and degradation of NMR-detectable compounds when the caffeine-containing coffee sample was roasted in the infrared and hot air roasters compared with the drum roaster. This can be attributed to the different types of heat transfer that occur in the roasters.

In addition, large differences in the formation and degradation of NMR-detectable compounds were observed between the caffeine-containing and decaffeinated coffee samples. Concentrations of kahweol, 5-CQA and trigonelline were significantly lower in the green coffee beans of the decaffeinated coffee sample compared with the caffeine-containing coffee sample. The degradation and formation of NMR-detectable compounds occurred at different times or to a lesser extent in the decaffeinated coffee sample compared with the caffeine-containing coffee sample. The observed differences between the caffeine-containing and decaffeinated coffee samples may have multiple explanations. One possible reason is a difference in geographical origin. However, it is probable that these disparities arose due to the decaffeination process partly eliminating other components from the green coffee beans, aside from caffeine. Also, the altered structure and surface of the coffee beans due to the decaffeination process could result in altered heat transfer. In order to assess the quality of the beans due to the decaffeination process and the use of different roasters, an additional sensory test should be conducted by trained assessors.

**Supplementary Materials:** The following supporting information can be downloaded from https://www.mdpi.com/article/10.3390/beverages9040087/s1. Table S1: Parameters for quantitation of the compounds in the coffee fat extracts using $^1$H-NMR spectroscopy and pulse-length-based concentration determination (PULCON). Table S2: Parameters for quantitation of the compounds in the aqueous coffee extracts using $^1$H-NMR spectroscopy and pulse length-based concentration determination (PULCON). 5-CQA, 5-caffeoylquinic acid; NMP, N-methylpyridine; HMF, 5-hydroxymethylfurfural. Figure S1: Evolution of (A) acetic acid and (B) 5-caffeoylquinic acid contents during the roasting process of the caffeine-containing coffee sample (C1) and decaffeinated coffee sample (C2) in the drum roaster (DR), infrared roaster (IR) and hot air roaster (HR). Figure S2: $^1$H-NMR spectra in the region of (A) $\delta_H$ = 6.2–8.0 ppm and (B) $\delta_H$ = 3.2–3.5 ppm of the aqueous extracts of the caffeine-containing coffee samples roasted in the (a) drum roaster, (b) infrared roaster and (c) hot air roaster recorded in $H_2O/D_2O$ (9:1, $v/v$) using a 400 MHz NMR spectrometer. Chemical shifts were calibrated using the TSP signal at $\delta_H$ = 0.00 ppm. For each roaster, the $^1$H-NMR spectra of the aqueous extracts of the samples taken every minute are displayed in multiple-layer mode. Signals of the protons of caffeine and aromatic protons of chlorogenic acid are highlighted in blue and orange.

**Author Contributions:** Conceptualization, D.W.L.; methodology, V.G., K.K. and D.W.L.; formal analysis, V.G. and K.K.; resources, D.W.L., E.W., P.W. and S.S.; data curation, V.G., K.K. and D.W.L.; writing—original draft preparation, V.G.; writing—review and editing, V.G., D.W.L., S.S., T.K., E.W., P.W. and K.K.; supervision, D.W.L. All authors have read and agreed to the published version of the manuscript.

**Funding:** This research received no external funding.

**Data Availability Statement:** The data presented in this study are available on request from the corresponding author.

**Acknowledgments:** Andreas Scharinger and Cedric Kunz are thanked for their excellent technical assistance.

**Conflicts of Interest:** S.S. is the owner of Coffee Consulate, Mannheim, Germany. Coffee Consulate is an independent training and research center. E.W. and P.W. work for the company that manufactures one of the coffee roasters under investigation. The other authors declare that they have no conflicts of interest.

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
