# Peer review of "Monitoring of Chemical Changes in Coffee Beans during the Roasting Process Using Different Roasting Technologies with Nuclear Magnetic Resonance Spectroscopy"

_beverages, doi:10.3390/beverages9040087_

Round 1

Reviewer 1 Report

ON the whole the paper was well written and the research conducted well. YOIu have several issues that need to be addressed, much int eh discussion as this was particularly poor and in some places missing.  There are 2 main issues with the paper. Because the coffees used for caffeinated and decaffeinated are from different origins, most of your results and conclusions about the roasting process affecting caffeinated versus decaffeinated cannot be made. At least not as presented. Since a noncaffeinated control of the decaffeinated coffee was not included nor was a decaffeinated version of the caffeinated coffee produced it is not possible to compare how roasting affected caffeinated versus decaffeinate since you do not know the original content of the proposed coffees. Many places the decaffeinated coffee showed lower amounts that the caffeinated but again it is unknown how much of this is due to the decaffeinated coffee or the roasting process. I recommend removing these comparisons and int he future ensuring you have included the correct controls. it is unclear why the authors didn't simply purchase 1 main coffee and then put half of it through the decaffeination process. Provide this justification in the manuscript would at least help with this poor design choice. 

While the authors do very briefly state about the compounds analyzed in this study in the introduction. It is not enough to clearly stated why these compounds are of interest. The results section would be greatly strengthened by adding an introductory sentence or 2 for each compound on the importance of these compounds to coffee quality. Do roasters want more of some and less than others. This is very unclear. 

What would make this paper even more applicable and of interest is if they provided some details about what of the compounds are desired for high quality coffee. Based on the different roasters would certain coffee be roasted shorter versus longer etc. While the authors have for the most part shown that NMR can be used, this application to roasters is lacking completely. 

You have several odd results based on comments. 1example being kahweol and is degradation product. It appears that the choice of analysis parameters does not allow for identification of both compounds. Why didn't you chose parameters where you could. You show that previous research have shown this degradation. You also describe how the decrease in furfuryl alcohol may be due to polymerization etc. Can you prove it from your data, as it appears there are inconsistent results based on other studies. Be careful when trying to explain what is happening. Particularly if you didn't measure the degradtion products. 

Author Response

ON the whole the paper was well written and the research conducted well. You have several issues that need to be addressed, much in the discussion as this was particularly poor and in some places missing.  There are 2 main issues with the paper. Because the coffees used for caffeinated and decaffeinated are from different origins, most of your results and conclusions about the roasting process affecting caffeinated versus decaffeinated cannot be made. At least not as presented. Since a noncaffeinated control of the decaffeinated coffee was not included nor was a decaffeinated version of the caffeinated coffee produced it is not possible to compare how roasting affected caffeinated versus decaffeinate since you do not know the original content of the proposed coffees. Many places the decaffeinated coffee showed lower amounts that the caffeinated but again it is unknown how much of this is due to the decaffeinated coffee or the roasting process. I recommend removing these comparisons and in the future ensuring you have included the correct controls. It is unclear why the authors didn't simply purchase 1 main coffee and then put half of it through the decaffeination process. Provide this justification in the manuscript would at least help with this poor design choice. 

Reply: Thank you for this comment to improve our manuscript: You are right. Since the caffeinated and decaffeinated coffee sample are from two different geographic origins, the results of the two coffee sample are not directly comparable. For example Consonni et al. described different amount of some coffee components depending on the geographic origin. Unfortunately, the magnitudes of these differences were not described. For the decaffeination process, the caffeine is extracted from the coffee beans by using water. Besides caffeine, other soluble coffee components were therefore also extracted. After caffeine was removed from the extraction solution, the coffee beans were allowed to reabsorb the water soluble components they lost during the extraction process. Due to this kind of extraction process, it is very likely that some compounds occur in smaller concentrations in the coffee beans after the decaffeination process.

To clarify this, we changed the following information:

Line 318-324: “However, the amount of kahweol in the decaffeinated coffee sample is considerably lower than the amount of kahweol in the caffeine-containing sample. This observation could be due to the decaffeination process. For the caffeine removal, water was used as extraction solvent. Water as solvent also dissolved other components from the coffee bean, which may not have been fully reabsorbed by the coffee beans after the extraction process.”

Line 342-350: “The results indicate a lower increase of furfuryl alcohol during the roasting process in the decaffeinated coffee sample than in the caffeine-containing coffee sample. Since furfuryl alcohol is formed during the coffee roasting process [35], the lower amount of this substance in the decaffeinated coffee indicates that the precursors occurred in lower amounts in the decaffeinated coffee sample. These differences could be due to the different geographical origin. However, some precursors could also been partially removed during the extraction in the decaffeination process.”

Line 405-408: “The different curves of formic acid which were observed in the caffeine-containing and decaffeinated coffee samples could be, among other reasons like the geographical origin, due to a different heat transfer.

Line 488-494: “Also, to be mentioned here, the caffeine-containing and decaffeinated coffee sample were from different geographical origins, which makes a direct comparison rather difficult. So varied amounts of 5-CQA in these two coffees could also be due to the different geographic origin. On the other hand, a significant amount of chlorogenic acid could be removed by the extraction solvent during the decaffeination process.”

Line 618-625: “The differences observed in the caffeine-containing and decaffeinated coffee sample could have several explanations. Thus, a different geographical origin of the caffeine-containing and decaffeinated coffee sample could be a reasons. However, it is likely, that these differences occur because the decaffeination process partially removes other components from the green coffee beans in addition to caffeine. Also, the altered structure and surface of the coffee beans due to the decaffeination process could result in altered heat transfer.”

While the authors do very briefly state about the compounds analyzed in this study in the introduction. It is not enough to clearly stated why these compounds are of interest. The results section would be greatly strengthened by adding an introductory sentence or 2 for each compound on the importance of these compounds to coffee quality. Do roasters want more of some and less than others. This is very unclear. 

Reply: Thank you for this note. We added some sentences in the introduction part to clarify the interest of the compounds investigated in this study.

Line 67-81: “Caffeine is one of the most widely consumed psychoactive substances [9]. Trigonelline and its thermal degradation product NMP are both nitrogenous compounds [9–11]. They have some beneficial effect on the cellular energy metabolism, chemopreventive acitivity and antioxidant activity [9–11]. Chlorogenic acids show anti-inflammatory and neuroprotective activity and are antioxidants, too [12]. Formic acid, acetic acid and lactic acid contribute among other to the sour taste of coffee. Studies indicate that the dierpenes cafestol and especially kahweol have antioxidant, antitumoral, chemoprotective and ant-inflammatory effects [13]. In contrast, HMF and furfuryl alcohol are components of concern in roasted coffee. The heat-induced contaminant furfuryl alcohol has been classified as a possible human carcinogen by the International Agency for Research and Cancer (IARC) [14], and studies have reported some evidence of carcinogenic activity of HMF in animal experiments [15].”

However, it is difficult to provide a clear connection of the compounds to the quality of the coffee. For sure, compounds of concerns should be reduced as much as possible during the coffee roasting process. On the other hand, the quality of a coffee is not only measured by the presence of beneficial components. For roasters, sensory analysis contributes a great part to the quality of a coffee. The sensory impression, in turn, is partly a relative and personal perception.

What would make this paper even more applicable and of interest is if they provided some details about what of the compounds are desired for high quality coffee. Based on the different roasters would certain coffee be roasted shorter versus longer etc. While the authors have for the most part shown that NMR can be used, this application to roasters is lacking completely. 

Reply: As mentioned above, since the quality of a coffee is also based on the sensory impression, providing a clear connection of compounds to high quality coffee is difficult here. In our studies, we focused on the analytical determination of coffee components without sensory analysis. We recognize the sensory analysis as an important contribution to the determination of the quality of the coffee beans and therefore pointed out the need of sensory analysis in the conclusion.

Line 625-627: “In order to assess the quality of the beans due to the decaffeination process and the use of different roasters, an additional sensory test should be conducted by trained assessors.”

You have several odd results based on comments. 1 example being kahweol and is degradation product. It appears that the choice of analysis parameters does not allow for identification of both compounds. Why didn't you chose parameters where you could. You show that previous research have shown this degradation. You also describe how the decrease in furfuryl alcohol may be due to polymerization etc. Can you prove it from your data, as it appears there are inconsistent results based on other studies. Be careful when trying to explain what is happening. Particularly if you didn't measure the degradtion products. 

Reply: Using 1H-NMR spectroscopy, determination of only dehydrokahweol is very difficult. To differentiate between kahweol and dehydrocahweol by means of 1H-NMR, only the protons at the double bond (H15 and H16) formed by water elimination can be used. However, this double bond also appears in the compound dehydrocafestol. The double bond of dehydrocahweol and dehydrocafestol formed by water elimination generates very similar 1H-NMR signals, so no differentiation can be made here. In order to achieve a differentiation of these compounds, further analytical methods such as liquid chromatography have to be used. According to Dias et al., there is no dehydrokahweol in green coffee beans (1). Pacetti et al. showed in their results that the content of dehydrokahweol after the roasting process is low compared to the content of kahweol. Therefore, this kind of analysis was sufficient for us (2).

(1) Dias, R.C.E., de Faria-Machado, A.F., Mercadante, A.Z. et al. Roasting process affects the profile of diterpenes in coffee. Eur. Food Res. Technol 239, 961–970 (2014).

(2) Pacetti, D.; Boselli, E.; Balzano, M.; Frega, N. G. Authentication of Italian Espresso coffee blends through the GC peak ratio between kahweol and 16-O-methylcafestol. Fod Chem. 135, 1569-1574 (2012).

Reviewer 2 Report

As mentioned by the authors of the reviewed paper, coffee is one of the most popular beverages in the world. And that popularity continues to grow. The taste and aroma of coffee depends on many factors. Authors used 1H nuclear magnetic resonance spectroscopy (NMR) to analyze fat and aqueous extracts of coffee beans roasted to different degrees of roasting using a professional drum roaster, a hot air fluidized bed sample roaster, and an infrared roaster. A caffeine-containing and a decaffeinated Coffea arabica coffee sample were used by Authors to monitor the roasting process in the different roasters.

The transformations of selected chemical compounds during coffee roasting were discussed in detail. The research was planned carefully, and the tools for its implementation were selected accordingly. The work from the scientific point of view is interesting and should find many recipients. I recommend the work for printing in Beverages.

One editorial note. Figure 3A needs correction in the description. It's in the legend DR_C1 twice, it should be DR_C2 the second time.

Author Response

As mentioned by the authors of the reviewed paper, coffee is one of the most popular beverages in the world. And that popularity continues to grow. The taste and aroma of coffee depends on many factors. Authors used 1H nuclear magnetic resonance spectroscopy (NMR) to analyze fat and aqueous extracts of coffee beans roasted to different degrees of roasting using a professional drum roaster, a hot air fluidized bed sample roaster, and an infrared roaster. A caffeine-containing and a decaffeinated Coffea arabica coffee sample were used by Authors to monitor the roasting process in the different roasters.

The transformations of selected chemical compounds during coffee roasting were discussed in detail. The research was planned carefully, and the tools for its implementation were selected accordingly. The work from the scientific point of view is interesting and should find many recipients. I recommend the work for printing in Beverages.

One editorial note. Figure 3A needs correction in the description. It's in the legend DR_C1 twice, it should be DR_C2 the second time.

Reply: Thank you for this note. We corrected the legend in figure 3A.

Reviewer 3 Report

The research article titled NMR Analysis of Coffee Roasting Processes provides valuable insights into the impact of various roasting methods on the chemical composition of coffee beans. This study specifically focuses on the utilization of 1H nuclear magnetic resonance spectroscopy (NMR) to analyze fat and aqueous extracts of coffee beans roasted using three different roasting techniques: a professional drum roaster, a hot air fluidized bed sample roaster, and an infrared roaster. Additionally, the research examines the differences in the roasting process between a caffeine-containing and a decaffeinated Coffea arabica coffee sample. This article is clear, well-written and concise in presentation of the research objectives, methods and findings. The discussion of moisture content changes during roasting is well-presented. The comparison between the caffeine-containing and decaffeinated coffee samples is particularly valuable, highlighting differences in moisture content evolution over time. This analysis sets the stage for understanding the subsequent changes in NMR-detectable compounds. The article effectively communicates the changes in NMR-detectable compounds during the roasting process, including forming and degrading various compounds like formic acid, acetic acid, lactic acid, 5-CQA, trigonelline, and NMP. These findings are crucial for understanding the chemical transformations that occur during coffee roasting and their potential impact on flavor and aroma. The mention of a time lag in the formation and degradation of NMR-detectable compounds in the caffeine-containing coffee sample roasted with the infrared and hot air roasters compared to the drum roaster highlights the influence of heat transfer mechanisms on the roasting process. This observation adds depth to the study.

Author Response

The research article titled NMR Analysis of Coffee Roasting Processes provides valuable insights into the impact of various roasting methods on the chemical composition of coffee beans. This study specifically focuses on the utilization of 1H nuclear magnetic resonance spectroscopy (NMR) to analyze fat and aqueous extracts of coffee beans roasted using three different roasting techniques: a professional drum roaster, a hot air fluidized bed sample roaster, and an infrared roaster. Additionally, the research examines the differences in the roasting process between a caffeine-containing and a decaffeinated Coffea arabica coffee sample. This article is clear, well-written and concise in presentation of the research objectives, methods and findings. The discussion of moisture content changes during roasting is well-presented.

The comparison between the caffeine-containing and decaffeinated coffee samples is particularly valuable, highlighting differences in moisture content evolution over time. This analysis sets the stage for understanding the subsequent changes in NMR-detectable compounds. The article effectively communicates the changes in NMR-detectable compounds during the roasting process, including forming and degrading various compounds like formic acid, acetic acid, lactic acid, 5-CQA, trigonelline, and NMP. These findings are crucial for understanding the chemical transformations that occur during coffee roasting and their potential impact on flavor and aroma. The mention of a time lag in the formation and degradation of NMR-detectable compounds in the caffeine-containing coffee sample roasted with the infrared and hot air roasters compared to the drum roaster highlights the influence of heat transfer mechanisms on the roasting process. This observation adds depth to the study.

Reply: Thank you for these comments regarding our manuscript.

Reviewer 4 Report

The authors have taken up the interesting topic of research on new methods of roasting coffee beans. It often happens that new methods, despite their many advantages, cause harmful effects, which is always better to verify. In the presented research, no drastic difference was observed.

For the enrichment of the reader's knowledge, a description of the method of decaffeination of coffee beans could be added, as well as more details of its origin (geographical coordinates of cultivation).

In the text, the time unit in Table 1 should be completed. It is described in the text but also should be putted in the Table description.

Author Response

The authors have taken up the interesting topic of research on new methods of roasting coffee beans. It often happens that new methods, despite their many advantages, cause harmful effects, which is always better to verify. In the presented research, no drastic difference was observed.

For the enrichment of the reader's knowledge, a description of the method of decaffeination of coffee beans could be added, as well as more details of its origin (geographical coordinates of cultivation).

Reply: Thank you for this note. You are right, a more detailed description of the decaffeination process would help to better understand the results obtained in this study. We added a more detailed description of the decaffeination process to the manuscript:

Line 120-127: “During this process, the coffee beans were threatened with water and steam to start the extraction process and males the beans expand. The coffee beans were then rinsed with water to extract the caffeine. This solvent also extract water soluble compounds, along with caffeine. In the next step, caffeine was removed from the solution using activated carbon. The caffeine beans were then allowed to reabsorb the molecules they lost during the extraction process.

In addition, we added more details of the geographical origin of the coffee beans:

Line 113-119: “The first Catuai (C1) was from Fazendas Dutra (São João do Manhuaçu, MG, Brazil, coordinates: 20°18’49.7”S; 42°07’33.9”W), where the coffee cherries were processed using the pulped natural method. The second Catuai (C2) was from Finca Hamburgo (Chiapas, Mexico, coordinates: 15°10’24.0”N; 92°19’46.1”W), where the coffee was processed using the fully washed method.”

In the text, the time unit in Table 1 should be completed. It is described in the text but also should be putted in the Table description.

Reply: Thank you for this note. We added the time unit [min] to the roasting time in Table 1.